# Towards a dynamic model to estimate evolving risk of major bleeding after percutaneous coronary intervention

Nathan C. Hurley[1‡], Nihar Desai[2,3‡], Sanket S. Dhruva[4], Rohan Khera[2,3], Wade Schulz[5], Chenxi Huang[2], Jeptha Curtis[2], Frederick Masoudi[6], John Rumsfeld[7], Sahand Negahban[8], Harlan M. Krumholz[2,3], Bobak J. Mortazavi[1,2,9]*

1 Department of Computer Science & Engineering, Texas A&M University, College Station, Texas, United States of America, 2 Center for Outcomes Research and Evaluation, Yale New Haven Health, New Haven, Connecticut, United States of America, 3 Section of Cardiovascular Medicine, Department of Internal Medicine, Yale School of Medicine, New Haven, Connecticut, United States of America, 4 University of California San Francisco, San Francisco, California, United States of America, 5 Laboratory Medicine, Yale School of Medicine, Yale New Haven Health, New Haven, Connecticut, United States of America, 6 Ascension, St Louis, Missouri, United States of America, 7 Department of Medicine (Cardiology), University of Colorado School of Medicine, Aurora, Colorado, United States of America, 8 Department of Statistics and Data Science, Yale University, New Haven, Connecticut, United States of America, 9 Center for Remote Health Technologies and Systems, Texas A&M University, College Station, Texas, United States of America

‡ These authors share first authorship on this work.
* bobakm@tamu.edu

## Abstract

While static risk models may identify key driving risk factors, the dynamic nature of risk requires up-to-date risk information to guide treatment decision making. Bleeding is a complication of percutaneous coronary intervention (PCI), and existing risk models produce only a single risk estimate anchored at a single point in time, despite the dynamic nature of this risk. Using data available from the National Cardiovascular Data Registry (NCDR) CathPCI, we trained 6 different tree-based machine learning models to estimate the risk of bleeding at key decision points: 1) choice of access site, 2) prescription of medication before PCI, and 3) choice of closure device. We began with 3,423,170 PCIs performed between July 2009 through April 2015. We included only index PCIs and removed anyone who had missing data regarding bleeding events or underwent coronary artery bypass grafting during the index admission. We included 2,868,808 PCIs; 2,314,446 (80.7%) before 2014 for training and 554,362 (19.3%) remaining for validation. This study considered all data available from the Registry prior to patient discharge: patient characteristics, coronary anatomy and lesion characterization, laboratory data, past medical history, anti-coagulation, stent type, and closure method categories. The primary outcome was any in-hospital bleeding event within 72 hours after the start of the PCI procedure. Discrimination improved from an area under the receiver operating characteristic curve (AUROC) of 0.812 using only presentation variables to 0.845 using all variables. Among 123,712

**Data availability statement:** NCDR data privacy policy means that the data cannot be made available. The protocol the ACC uses to collect the NCDR Registry data from hospitals restricts access to the data. Access to the data can be requested from the ACC/NCDR by proposing additional work on the data here: https://cvquality.acc.org/NCDR-Home/research/submit-a-proposal. Once access to Cath-PCI is granted, the following code will allow readers to replicate the work. Code is available here: https://github.com/stmilab/DynamicBleeding/.

**Funding:** The author(s) received no specific funding for this work.

**Competing interests:** I have read the journal's policy and the authors of this manuscript have the following competing interests: Dr Desai reported receiving grants and personal fees from Amgen, Boehringer Ingelheim, and Cytokinetics and personal fees from Relypsa, Novartis, Medicines Company, and SC Pharmaceuticals outside the submitted work. Dr. Dhruva receives funding from the Department of Veterans Affairs and Arnold Ventures. Dr. Masoudi serves on a study steering committee for the American College of Cardiology and has previously served as the Chief Scientific Advisor of the ACC NCDR. Dr. Rumsfeld is employed by Meta Platforms, Inc., but this work was completed prior to that employment. Dr Mortazavi reported receiving grants from the National Science Foundation, the National Institute of Biomedical Imaging and Bioengineering, National Heart, Lung, and Blood Institute, US Food and Drug Administration, and the US Department of Defense Advanced Research Projects Agency outside the submitted work; in addition, Dr Mortazavi has a pending patent (US20180315507A1) and is a consultant for Hugo Health and for Ensight-AI. In the past three years, Harlan Krumholz received options for Element Science and Identifeye and payments from F-Prime for advisory roles. He is a co-founder of and holds equity in Hugo Health, Refactor Health, and ENSIGHT-AI. He is associated with research contracts through Yale University from Janssen, Kenvue, and Pfizer. Dr. Khera receives support from the National Heart, Lung, and Blood Institute of the National Institutes of Health under awards 1K23HL153775 and 1R01HL167858-01A1,

patients classified as low risk by the initial model, 14,441 were reclassified as moderate risk (1.4% experienced bleeds), while 723 were reclassified as high risk (12.5% experienced bleeds). Static risk prediction models have more predictive error than those that update risk prediction with newly available data, which provides up-to-date risk prediction for individualized care throughout a hospitalization.

## Author summary

Clinical risk models used for treatment decision making are often static models used with fixed input at a fixed point of time. Risk of adverse events, however, is dynamic, changing throughout admissions because of treatment decision making. This work looks at the risk of major bleeding for patients undergoing percutaneous coronary intervention, showing the changes in patient risk estimation throughout the course of treatment. By identifying the changes in risk of bleeding at different points in time, we demonstrate the need for more dynamic evaluation of risk estimates, providing potential changes in treatment decision making throughout admissions, accounting for prior treatment decisions made. The models demonstrate an improvement in discrimination in predicting risk of major bleeding and demonstrates a reclassification of a subset of patients, particularly demonstrating the need for re-evaluating bleeding risk (and thus treatment with bleeding avoidance therapies) at various stages of patient admission before discharge. Models that update risk prediction with newly available data, which provides up-to-date risk prediction, enable individualized care throughout a hospitalization.

## Introduction

Bleeding is a common complication of percutaneous coronary intervention (PCI), leading to significant morbidity, mortality, and cost [1]. Several tools have been developed to predict post-PCI bleeding, including two models [2–4] from the American College of Cardiology's National Cardiovascular Data Registry (NCDR) [5]. By informing clinicians about bleeding risk, these models can aid the selection of bleeding avoidance strategies, such as combination of radial access and the use of bivalirudin interprocedurally as well as post-procedural care strategies, particularly in high-risk patients, thereby reducing rates of major bleeding complications and improving care quality and clinical outcomes [6,7]. A problem with existing models is that they produce a single estimate of bleeding risk anchored at a single point in time, i.e., data prior to the procedure inform a single estimate for risk of post-PCI bleeding. These models do not update the risk estimates as new clinical information emerges. Therefore, as treatment decisions are made or unforeseen events occur, these models are unable to adapt and incorporate new information.

Risks are dynamic in nature; as data are gathered and treatment decisions are made, risk estimates should account for all the information currently available,

is the coinventor of U.S. Provisional Patent Application No. 63/177,117, "Methods for neighborhood phenomapping for clinical trials", and is a founder of Evidence2Health and Ensight-AI. Dr. Schulz is a technical consultant to Hugo Health, a personal health information platform, and co-founder of Refactor Health, an AI-augmented data management platform for healthcare; is a consultant for Interpace Diagnostics Group, a molecular diagnostics company. These research data were provided by the American College of Cardiology's National Cardiovascular Data Registry, Washington, DC. This study was funded in part by the American College of Cardiology Foundation. The remaining authors have nothing to disclose.

including changes in clinical status of patients as a response to treatment decisions [8]. Dynamic models can enable estimation of risk that adapts and updates throughout an episode of care. For example, in the ACUITY trial that evaluated the use of bivalirudin in acute coronary syndrome, intra-procedural events were strongly associated with adverse outcomes [9], which would be expected to alter antiplatelet and anticoagulant strategies. We hypothesize that bleeding risk is similarly a dynamic process, affected by multiple pre-, intra-, and post-PCI patient and procedural factors throughout the care pathway.

Risk models that update across the patient episode of care have the potential to improve our ability to individualize risk prediction; static risk prediction models may falter over time, missing important predictors that become available throughout admission and treatment. Providing physicians up-to-date feedback may inform optimization of therapeutic strategies through enhanced decision-support at actionable points during an episode of care involving a PCI procedure. These models may also improve the understanding of the dynamics and key variables affecting bleeding risk, representing a transformational change in risk prediction and embrace the principles of a learning health care system [10]. Accordingly, we sought to develop models that update estimates of patient risk of bleeding over time, enabling a dynamic estimate of risk that incorporates evolving clinical information. Taking the variables from the prior published risk models that used logistic regression and boosted decision trees from a wide compliment of available variables [3,4], we sought to demonstrate that risk stratification changes over time and notion of available data. In addition, we then sought to define a parsimonious model, with a number of colinear factors removed in order to compare findings.

## Results

### Patient cohort, variables used, and overall performance

We included 2,868,808 PCIs in the NCDR CathPCI registry; 2,314,446 (80.7%) prior to 2014 for model training and 554,362 (19.3%) after 2014 for validation and model interpretation. The mean (SD) age of patients was 64.6 (12.0) years and 68.3% were male (Table 1). Overall, there were 118,327 (4.1%) major bleeding events: 98,167 (4.2%) major bleeding events in the training set and 20,160 (3.6%) in the validation set.

Models were trained in a stratified, five-fold cross validation. Model performance metrics at each stage are provided in Table 2, and each stage is described further below. The model area under the receiver operating characteristic curves (AUROCs) increase from 0.812 to 0.845, and model area under the precision recall curves (AUPRCs) increase from 0.203 to 0.242, respectively. These results included decision tree models that use all available data for risk stratification. S1 Table lists all the variables used in our XGBoost-trained model. We maintained a number of extracted features persistent across prior models by Rao et al.[3] and Mortazavi et al.[4], in order to demonstrate that all the available variables, when considered in a staged setting, allow further clarity towards the dynamic nature of Risk of Major Bleeding. We also sought to evaluate if the multicollinearity of these variables impacted the modeling, as a separate experiment, in order to clearly identify performance gains as a result

**Table 1. Patient characteristics.**

| | Overall | Training | Validation |
|---|---|---|---|
| | (n = 2,868,808) | (n = 2,314,446) | (n = 554,362) |
| Demographics | | | |
| Age, mean (SD), y | 64.6 (12.0) | 64.6 (12.0) | 64.9 (11.9) |
| Men | 1,960,409 (68.3%) | 1,577,369 (68.2%) | 383,040 (69.1%) |
| BMI, mean (SD) | 30.0 (6.4) | 30.0 (6.4) | 30.1 (6.4) |
| Cardiovascular Comorbidities | | | |
| Diabetes | 1,057,221 (36.9%) | 844,928 (36.5%) | 212,291 (38.3%) |
| Hypertension | 2,353,798 (82.1%) | 1,895,949 (81.9%) | 457,849 (82.6%) |
| Peripheral Vascular Disease | 339,316 (11.8%) | 274,039 (11.8%) | 65,278 (11.8%) |
| Chronic Kidney Disease | 861,391 (30.0%) | 705,765 (30.5%) | 155,626 (28.1%) |
| Previous PCI | 1,178,346 (41.1%) | 948,367 (41.0%) | 229,978 (41.5%) |
| Previous CABG | 510,781 (17.8%) | 414,560 (17.9%) | 96,222 (17.4%) |
| PCI Procedural Status | | | |
| Elective | 1,196,485 (41.7%) | 992,525 (42.9%) | 203,961 (36.8%) |
| Urgent | 1,152,328 (40.2%) | 906,226 (39.2%) | 246,101 (44.4%) |
| Emergent | 512,404 (17.9%) | 409,659 (17.7%) | 102,744 (18.5%) |
| Salvage | 6,440 (0.2%) | 5,029 (0.2%) | 1,411 (0.3%) |
| Unknown | 1,150 (0.04%) | 1,007 (0.04%) | 145 (0.03%) |
| STEMI | 468,270 (16.3%) | 373,792 (16.2%) | 94,477 (17.0%) |
| Cardiogenic Shock | 64,743 (2.3%) | 51,689 (2.2%) | 13,055 (2.4%) |
| Cardiac arrest within 24h of PCI | 49,008 (1.7%) | 38,840 (1.7%) | 10,168 (1.8%) |
| Preprocedural hemoglobin, median (IQR), g/dL | 13.7 (12.4-14.9) | 13.7 (12.4-14.9) | 13.7 (12.4-14.9) |
| Access Site | | | |
| Femoral | 2,394,173 (83.5%) | 1,997,049 (86.3%) | 397,124 (71.6%) |
| Radial | 474,635 (16.5%) | 317,397 (13.7%) | 157,238 (28.4%) |
| Medications Used | | | |
| Ticlopidine | 5,895 (0.2%) | 5,186 (0.2%) | 709 (0.1%) |
| Clopidogrel | 1,988,178 (69.3%) | 1,654,031 (71.5%) | 334,147 (60.3%) |
| Prasugrel | 433,079 (15.1%) | 339,902 (14.7%) | 93,177 (16.8%) |
| Ticagrelor | 167,838 (5.9%) | 80,318 (3.5%) | 87,520 (15.8%) |
| Fondaparinux | 15,816 (0.6%) | 14,837 (0.6%) | 979 (0.2%) |
| Low Molecular Weight Heparin | 272,261 (9.5%) | 224,180 (9.7%) | 48,081 (8.7%) |
| Unfractionated Heparin | 1,528,882 (53.3%) | 1,197,304 (51.7%) | 331,578 (59.8%) |
| Bivalirudin | 1,695,225 (59.1%) | 1,375,031 (59.4%) | 320,194 (57.8%) |
| GP llb/llla | 677,865 (23.6%) | 576,753 (24.9%) | 101,112 (18.2%) |
| Direct Thrombin Inhibitor | 29,512 (1.0%) | 24,961 (1.1%) | 4,551 (0.8%) |
| Closure Method | | | |
| Manual Compression | 965,618 (33.7%) | 815,354 (35.2%) | 150,264 (27.1%) |
| Sealant | 916,374 (31.9%) | 745,456 (32.2%) | 170,918 (30.8%) |
| Mechanical | 512,968 (17.9%) | 358,927 (15.5%) | 154,041 (27.8%) |
| Suture | 264,494 (9.2%) | 215,420 (9.3%) | 49,074 (8.9%) |
| Patch | 99,690 (3.5%) | 84,853 (3.7%) | 14,837 (2.7%) |
| Staple | 184 (0.0%) | 171 (0.0%) | 13 (0.0%) |
| Other | 94,818 (3.3%) | 82,626 (3.6%) | 12,192 (2.2%) |
| None/Missing | 14,662 (0.5%) | 11,639 (0.5%) | 3,023 (0.5%) |
| Post-PCI Major Bleeds | 118,327 (4.1%) | 98,167 (4.2%) | 20,160 (3.6%) |

**Table 2. Comparison of model performances for bleeding prediction.** We compare model performance by area under the receiver operating characteristic curve (AUROC), the area under the precision recall curve (AUPRC), the Brier Skill Score, and the Reliability and Resolution of the Brier Decomposition. Brier Skill Score and Resolution are higher for better performance, reliability should be lower for better performance.

| Model | AUROC | AUPRC | Brier Skill Score | Brier Reliability | Brier Resolution |
|---|---|---|---|---|---|
| **Presentation** | 0.812 (0.812-0.812) | 0.203 (0.203-0.203) | 0.088 (0.088-0.088) | 2.6E-4 (2.6E-4-2.6E-4) | 3.3E-3 (3.3E-3-3.3E-3) |
| **+ Access Site** | 0.817 (0.817-0.817) | 0.204 (0.204-0.205) | 0.091 (0.091-0.091) | 2.0E-4 (1.9E-4-2.0E-4) | 3.4E-3 (3.4E-3-3.4E-3) |
| **+ Cath Lab** | 0.825 (0.825-0.825) | 0.208 (0.208-0.208) | 0.094 (0.094-0.094) | 2.1E-4 (2.0E-4-2.1E-4) | 3.5E-3 (3.5E-3-3.5E-3) |
| **+ Medication Decision** | 0.832 (0.832-0.832) | 0.217 (0.216-0.217) | 0.102 (0.102-0.102) | 1.4E-4 (1.3E-4-1.4E-4) | 3.7E-3 (3.7E-3-3.7E-3) |
| **+ PCI Variables** | 0.844 (0.844-0.845) | 0.241 (0.240-0.241) | 0.118 (0.118-0.118) | 1.1E-4 (1.1E-4-1.2E-4) | 4.2E-3 (4.2E-3-4.2E-3) |
| **+ Closure Decision** | 0.845 (0.845-0.845) | 0.242 (0.241-0.242) | 0.119 (0.119-0.119) | 1.0E-4 (1.0E-4-1.1E-4) | 4.3E-3 (4.3E-3-4.3E-3) |

of variables versus performance gains as a result of the dynamic staging of models. By removing all the created, binary indicator features, leaving only their continuous or categorical variables from which they were extracted, we actually saw a slight decrease in model performance, with AUROCs ranging from 0.808 to 0.843 and AUPRCs from 0.198 to 0.237. Because of the way the gradient boosted decision trees model each weak learner across the participants in the training dataset, it makes sense that the non-linear risk is better modeled with a wider array of features, some continuous, some categorical, and some extracted binary daughter variables. Further detail of this additional experiment is described in the S1 Text.

The respective receiver operating characteristic plots and precision-recall curve plots for each stage of the model are presented in S1 and S2 Figs, respectively (and S3 and S4 Figs for the multicollinearity analysis). For each of these figures, a single fold from the five-fold cross-validation was selected for plotting, with the confidence intervals in Table 2 demonstrating confidence in an accurate illustration.

The staging of models is designed to be aligned with key clinical decision-making points. However, by the definition of major bleeding, it is possible that these are confounded with periprocedural bleeding rather than bleeding that can take place up until 72 hours after the procedure. As a result, the final two models can only be used as a decision-making model if major bleeding has not yet occurred. Finally, discussed further in the S2 Text (and in S2 Table), we found that the gains in performance and risk stratification were consistent across Femoral and Radial subgroups.

## Interpreting SHAP plots

One approach for interpreting models is SHapley Additive exPlanations (SHAP) [11]. SHAP feature importance plots for the Presentation Model and the Closure Model are in Fig 1 (with intermediate models shown in S5–S8 Figs and SHAP plots with the colinear variables removed in S9–S14 Figs). In each plot, the variables are sorted in order of decreasing importance. The color of a variable relates to the value of that variable, while location along the x-axis represents how much that variable contributes to the risk of bleeding. Features most strongly driving predictions of high bleeding risk appear on the right with high SHAP values, while features most predictive of low bleeding risk appear at the left. The specific plots, per stage, are described further below. Note that the plots do not show the interactions across the depth of the tree but present a broader view of overall risk per feature.

## Stage 1: Clinical presentation (Model 1)

The initial model uses information available to a clinician at the time that a patient presents and predicted bleeding risk with an AUROC of 0.812 and AUPRC of 0.203. The Brier skill score of this model is 0.088, representing the degree that this model improves over a naïve model (higher is better). The Brier reliability is 2.6E-4, representing distance to true probabilities (lower is better), while the resolution is 3.3E-3, representing forecast distances to the mean rate (higher is

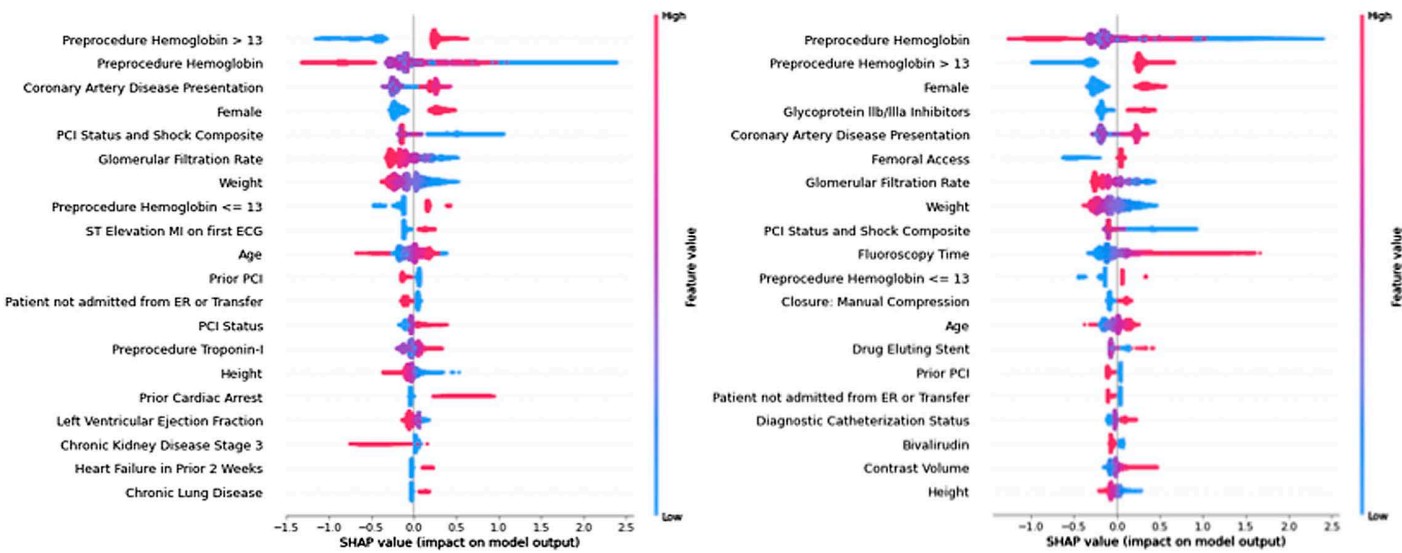

**Fig 1. SHAP Tree explainer for (a) Model 1: Presentation and (b) Model 6: Closures.** Variables are sorted in order of decreasing importance. Variable color relates to the value of that variable, while location along the x-axis represents how much that variable contributes to the risk of bleeding. Features most strongly driving predictions of high bleeding risk appear on the right, while features most predictive of low bleeding risk appear at the left. The higher a variable is, the greater overall importance that variable exhibits for the model. Red values indicate high values for that variable (if continuous or ordinal) or "true" (if binary), while blue values indicate the opposite. Points to the right of the axis (positive SHAP values) indicate that a feature of that value increases model estimate of bleeding risk, while points to the left of the axis (negative SHAP values) indicate that a feature of that value decreases the model estimate of bleeding risk. For instance, on the top row of Fig 2, a preprocedural hemoglobin greater than 13 is associated with an increased risk of bleeding, while value less than 13 is associated with a decreased risk. The wide range of SHAP values for this variable shows that while the direction of association is constant, the degree to which this feature impacts risk is not constant.

better). The SHAP plot (Fig 1A) shows that this model is most strongly driven by pre-procedural hemoglobin and coronary artery disease symptoms at presentation. The inclusion of both continuous variables and dichotomized show the importance of different values of this variable across decision trees in the final model. Variable nonlinearity is also demonstrated here. Looking at the top two rows of the figure, hemoglobin > 13 is associated with an increased risk, but the continuous hemoglobin variable shows that very low hemoglobin is more strongly associated with bleeding, while a relatively higher hemoglobin is associated with a lower risk of bleeding, but a wider array of increases in risk across the intermediate values just above the threshold make the dichotomous variable represent increased risk rather than more varied risk at different thresholds.

### Intermediate decision points and variables

A complete discussion of the model performance, variable importance, and patient reclassification of each intermediate stage are available in the S3 Text. In brief, the model AUROCs improve to 0.817, 0.825, 0.832, and 0.844 with variable importance changing as additional data becomes available.

### Decision 3: Closure method (Model 6)

The final decision point is closure. This decision had minimal effect on overall prediction, with AUROC remaining at 0.845 and AUPRC improving slightly to 0.242. The Brier skill score improved slightly to 0.119, the Brier reliability improved to 1.0E-4, and the Brier resolution improved to 4.3E-3. Fig 1B shows a SHAP explanatory plot for this model. Manual compression is the closure method most strongly predictive of increased bleeding risk (12[th] most informative variable).

## Reclassification from Model 1 to Model 6

Total reclassification from the initial model to the final model is shown in Table 3. Among 123,712 patients classified as low risk by the initial model, 14,441 (11.7%) were reclassified as moderate risk, while 723 (0.6%) patients were reclassified as high risk. Among the 14,441 patients re-classified as moderate risk, 1.4% experienced bleeding events. Among those 723 patients reclassified to high risk, 12.5% experienced bleeding events. Among 270,485 patients classified as moderate risk by the initial model, 82,418 (30.5%) were reclassified to low risk by the final model, while 16,577 (6.1%) were reclassified to high risk by the final model. Among the 82,418 patients reclassified to low risk 0.5% experienced bleeding events, while among the 16,577 patients reclassified to high risk 7.0% experienced bleeding events. Finally, among 160,165 patients classified as high risk by the initial model, there were 40 (<0.1%) patients reclassified to low risk, and 43,265 (27.0%) patients reclassified to moderate risk. The 40 patients reclassified to low risk experienced no bleeds (bleeding rate of 0%), while 2.5% of the 43,265 patients reclassified to moderate risk had a bleeding event. Intermediate reclassification results are available in the S3 Text.

## Discussion

In a large national registry, we demonstrate that the risk of post-PCI bleeding is dynamic and changes with treatment decisions. As additional data are gathered over time, estimates of this risk are meaningfully updated. We also found that as more data becomes progressively available, the model better identifies associations between variables and subsequent bleeding. The staged nature better represents an individual's risk throughout the course of treatment and can inform treatment decisions in a way that is superior to using a pre-PCI model in isolation. As the model obtains more data, the association of this and the risk of bleeding increases in importance, as the decision itself represents a modest increase in risk at Model 2, but a larger increase in risk when additional data representing the case becomes apparent.

By extending beyond prior static machine learning research, which demonstrated that the full dynamic range of available variables allows higher-order, non-linear models to improve estimates of risk (4), this work demonstrates how the association of these variables in in all relevant formats (continuous and dichotomous) change in representation of risk throughout the course of care. Bleeding avoidance strategies were less used in those at highest risk based upon static models [12]. While patients would receive only therapies with lower risk of bleeding, in some cases this must be balanced by antiplatelet and anticoagulant strategies that may reduce risk of coronary ischemia but with potential bleeding risk. Decision support that balances these factors may help to, therefore, inform other clinical decisions such as the PCI

**Table 3. Shift table across models.** Top value in each cell is number of patients classified into that risk bin by the two respective models. The bottom value in each cell indicates the actual bleeding rate of all patients within that cell.

**Initial vs Final Estimate**

| Closure Decision Model | Model 1 | | | |
| | <1% | 1-4% | >4% | All |
| | Patients, N | Patients, N | Patients, N | Patients, N |
| | Observed Rate | Observed Rate | Observed Rate | Observed Rate |
| <1% | 108,548 | 82,418 | 40 | 191,006 |
| | 0.41% | 0.48% | 0.00% | 0.44% |
| 1-4% | 14,441 | 171,490 | 43,265 | 229,196 |
| | 1.47% | 1.59% | 2.50% | 1.76% |
| >4% | 723 | 16,577 | 116,860 | 134,160 |
| | 12.45% | 6.99% | 12.02% | 11.40% |
| All | 123,712 | 270,485 | 160,165 | 554,362 |
| | 0.60% | 1.58% | 9.45% | 3.64% |

procedure itself (e.g., extent of revascularization performed may be more complete if a patient is at lower bleeding risk during the study).

To date, studies have failed to leverage the large numbers of variables tracked over an episode of care [8]. Prior efforts to update models that estimate the risk of bleeding via machine learning still restricted utility of the model to a single decision point in time, using only the available data for when that decision would be made [4]. This work identifies the dynamic nature of this bleeding risk and emphasizes the need for learning representations of risk over time. Additionally, this work identifies the utility of variables that may not have been appreciated to have an association with bleeding. The findings suggest the potential for additional improvement through direct integration with the electronic health record. By accessing data in near real-time, models will be able to present individualized estimates of risk and evaluation of treatment decisions, personalizing decision-making and care throughout hospitalization and PCI.

This work enables future prospective modeling in electronic health record-native implementation. While we segmented the data on key decision points within the registry data, we demonstrate the potential for implementing a clinical decision making tool within the medical record for real-time use, where each available data point can be added for key stages within each hospital system's clinical workflow. This would potentially benefit different clinical settings that have varied anti-coagulation efforts, ensure that the risk/treatment paradox with anti-coagulation is addressed, where those at highest risk are not being treated [13]. This would provide the potential for biggest impact, identifying cases in real time that should or should not have bleeding avoidance therapies to help make continuous decisions in the best interest of patient outcomes.

## Limitations and future directions

There are several key limitations and opportunities for future research in our study. First, real-time data, which would be necessary for proactive implementation of dynamic models, can be challenging to acquire. The NCDR relies upon manual chart abstraction for many variables, such as past medical history variables and as a result, data we have at our disposal is older, and at risk for data drift issues with respect to changes in clinical practice. Our best efforts to adjust for this was the temporal split. The implementation of such a system at scale within an electronic health record environment, where data may be available within near real-time capacity, requires additional investigation of natural language processing techniques, models that handle advanced time-series data, and appropriate evaluation of user interface design for reducing clinician burden when interacting with such a model. This, additionally, would require an evaluation of the quality of the features being fed into the model to ensure those risk factors identified in this work are easily available and readily of high quality without the rigorous review the registry data undergoes.

The second is the timing of the available variables. The registry abstracts some variables as pre- and intra- procedure, so some confounding may exist from reactions to bleeds during the procedure. This nature of the variables could prevent accounting for some intra-procedural variables, such as if there was initially acute stent closure after PCI, which could require additional steps in the procedure. Additionally, the lack of timing prevents us from demonstrating model degradation as a function of time. It can only be inferred through the use of additional variables here. That said, we still find same estimation improvements and risk changes through the addition of medication, which we believe supports our conclusions for needing a dynamic model.

Finally, the definition of the bleeding outcome includes multiple items in a composite form, which includes a hemoglobin cutoff. This suggests that patients with a low baseline hemoglobin were more likely to experience this outcome, because a small drop would put them below the threshold as opposed to a substantial drop being needed for other patients.

## Conclusion

We have developed a model that provides dynamically updated bleeding risk assessments by incorporating information available at different stages of patient care among patients undergoing PCI. These methods demonstrate evolution in variable importance as clinical decisions are made through course of care. For the risk of in-hospital bleeding, variable

importance and bleeding risk changes as variables are included from cardiac catheterization and PCI. Accounting for the time-varying nature of data and capturing the association between treatment decisions and changes in risk provide up-to-date information that may guide individualized care throughout a hospitalization.

## Materials and methods

### Study cohort

This study included all index PCIs in the National Cardiovascular Data Registry (NCDR) CathPCI registry version 4.4 from July 2009 through April 2015, which contains patient data from over 1500 sites across the United States [5,14]. To examine the improvement of dynamic data over static risk prediction models, we used the same cohort as Mortazavi et al. [4]. Therefore, we excluded patients during readmissions, who died in the hospital, who had missing data regarding if they had any bleeding events, or who underwent coronary artery bypass grafting (CABG) during the index admission [3,4].

The primary outcome was any in-hospital bleeding event within 72 hours after the start of the PCI procedure. Bleeding was defined using the definition employed in prior NCDR risk models as a hemoglobin drop ≥3 g/dL, whole blood or packed red blood cell transfusion, or intervention/surgery at the bleeding site to reverse/stop or correct bleeding. We further excluded patients with multiple, unknown, or brachial access sites to evaluate the treatment decision point of radial versus femoral access. We additionally excluded patients with multiple closure methods. This final exclusion was added because multiple closure methods may be associated with multiple access or may represent bleeding (e.g., a femoral closure device did not work and so manual pressure was held), which would bias the model to predicting risk of bleeding after the bleed has occurred.

### Variables of interest

This study considered all data available from the CathPCI Registry prior to patient discharge [5]. This included all data used by the existing models [3,4], as well as additional variables as described below. The current full existing NCDR bleeding risk model [3] uses 31 variables: 23 patient characteristics at the time of presentation and 8 characteristics related to coronary anatomy and lesion characterization (see S4 Text). A more recent work added additional variables from the registry related to those 31 variables, finding improved predictive performance (4). Our study includes further additional data. The additional data considered in this study consists of additional laboratory data, past medical history, additional coronary anatomy (including percent stenosis), stent type, and closure method categories (manual compression, sealant, mechanical, suture, patch, staple, other, or none).

We recognize that the variables have multicollinearity, particularly the derived indicator features based upon the continuous feature counterparts. As this work sought to demonstrate changes in risk over time, we did not remove any variables used in the prior two studies, as we sought to only change the time and addition of new variables at each stage in evaluation. As a result, we acknowledge some co-linearity of variables as a potential limitation in performance and in interpretation. As a result, the S1 Text also discusses the same conducted analysis with a reduced feature set, where those particular indicator features have been removed, and we find that performance is comparable, though slightly stronger when including all possible predictors for the model to explore the decision space with.

### Staged model analysis

We sorted all available CathPCI data into key decision stages of a PCI episode of care. First, we defined three decision points that affect bleeding risk: 1) choice of access site (radial versus femoral); 2) choice of medications (including those administered 24 hours pre-procedure and intra-procedure); and 3) choice of closure device indicator variables. While the choice of closure for radial access would be expected to be none, there are closure devices used with Radial Access. Without an ability to do a chart review, we take this data as is (and discuss further in the S4 Text).

Using these three key decision points, we evaluated variables available at three stages: 1) variables available at patient presentation prior to PCI; 2) variables available after diagnostic coronary angiography; and 3) variables related to the PCI procedure. Combining these three decision points and the information available at each of them, six models were designed (Fig 2). The first included only variables available at presentation to the cardiac catheterization lab. Each subsequent model added either a decision node or information that could inform the next decision. The final model included all variables and clinical decisions through the PCI procedure, evaluating remaining bleeding risk for post-procedural care. The final details of the variables and which stage they are provided to are provided in the S4 Text. Knowing that radial and femoral access choice lead to different findings, we also conducted a subgroup analysis in which we evaluated model performance on each subgroup, discussed further in the S2 Text.

### Data preparation

The NCDR utilizes a high-quality review and adjudication process, ensuring minimal missing data across variables [15]. Additionally, several steps in data preparation were necessary prior to model development, including data cleaning to determine the true rate of missing, including for daughter variables only collected if the indicator variable suggests it should have. Details are provided in the S4 Text.

Once data was cleaned, the missing values were imputed using multiple multivariate feature imputation. Each missing feature was modeled using Bayesian ridge regressors trained in a round-robin fashion. This imputation was performed using the Iterative Imputer package in scikit-learn 0.24.1 [16], based on the multivariate imputation by chained equations

**Fig 2. Model hierarchy.** Each model integrated information of all features from prior models, as well as an added set of features. Passage through an episode of care proceeds from the top to the bottom. Case study risks over time at each stage are shown here.

PLOS Digital Health

(mice) package for R [17]. Following imputation, binary and ordinal variables were set to the nearest allowed value. Multiple imputations were produced by sampling from the regressor models multiple times; each discrete sampling was a new overall sample from the model. This sampling was used to produce five folds of imputations. We used a re-imputation technique for handling the longitudinal nature of the data, producing multiple imputed datasets at each stage through an episode of care [18]. The imputation models were trained on the training dataset prior to cross validation [19]. The test sets were multiply imputed, but the regressors used for this imputation were trained only on training data to avoid data leakage. A positive case was one in which a major bleed occurred, a negative case when no major bleed occurred.

## Training, testing, and evaluating

The cohort was divided into temporal subsets: an initial 80% subset (July 1, 2009 through December 31, 2013) for model training and a later 20% subset (January 1, 2014 through December 31, 2014) for model validation [20,21]. This temporal split allows for the fairest possible evaluation of the model through testing it on the most distinct subset. This approach protects against biasing the result of the overall model by including more recent data, as in other recent NCDR models [22]. While an external validation set would be preferable, this method of separating data allows for the most realistic testing of the model (testing models trained on retrospective data on newly-seen patients). Patient characteristics for both the entire dataset as well as the training and validation splits are shown in Table 1.

As discussed above, five imputation folds were created for each stage. The model used was XGBoost, a gradient descent boosted decision tree model [23], as this model has demonstrated improved performance over linear classifiers [4]; because prior work demonstrated the improvement in model performance of XGBoost over linear classifiers (logistic regression), this work focuses only on the improvement in the XGBoost model over the stages of analysis [4]. We conducted an internal five-fold cross-validation to tune the hyperparameters of the model for each stage and each fold. The hyperparameters tuned were maximum depth of each tree (2, 4, 6, or 8), number of tree estimators (100, 500, 1000, or 5000), and learning rate for the model (0.1, 0.15, 0.2, or 0.3). This internal cross-validation was performed using a halving grid search as implemented by the scikit-learn package HalvingGridSearchCV [16]. Optimal performance was found in 27 of the 30 experiments (6 stages * 5 folds) to be given by 1000 estimators with a max depth of 2 and a learning rate of 0.1.

The five imputation folds allow for training and validating the model multiple times, providing both estimates of overall model performance and uncertainty [20]. From this, we calculated the area under the receiver operating characteristics curve (AUROC) for evaluating model discrimination (c-statistic). The AUROC, alone, may be insufficient to properly describe model performance, particularly in the presence of imbalanced event rates, therefore we supplement this measure with additional measures of predictive performance [24]. To better understand positive predictive value across the full range of risk stratification, we also calculated the area under the precision recall curve (AUPRC). Briefly, the precision-recall curve calculates the tradeoff between precision (positive predictive value) and recall (sensitivity) across the full range of thresholds [25]. This model calibration also allows calculation of the Brier Skill Score, which provides an assessment of model calibration on an easy to interpret scale of 0–100% for calibration fit, as well as the Brier Decomposition (Reliability and Resolution), which provides measures of over and under estimation of the calibration curve.

## Variable importance

Beyond model performance, model interpretation is a key factor in clinical utility [20]. One approach for interpreting models is SHapley Additive exPlanations (SHAP) [11]. SHAP attributes an importance value to each feature of a given set, therefore allowing for an ordering of features from greatest to least impact on model output. SHAP values were generated to provide visual understanding about the impact of factors driving changes in accuracy of the risk prediction model and decisions through the stages outlined in Fig 1.

To further understand dynamic risk predictions following a decision, shift tables were generated. For these, patients were classified into categories of low risk (<1%), moderate risk (1%-4%), or high risk (>4%) of bleeding, based upon

the training set overall event rate of 4.2% and empirically divided to approximately balance patients between categories across all models [22]. These shift tables are useful for visualizing changing risks of bleeding before and after a decision or for comparing the performance of the initial and final models. Shift tables were selected over confusion matrices because a confusion matrix would require setting specific thresholds for a positive or negative prediction; shift tables are a more sophisticated approach to demonstrating confusion matrix results by demonstrating changes in predicted probability without pre-selecting a specific decision threshold.

### Case studies

To further understand dynamic risk and changes in variable performance, we then cherry picked two example cases that had changes in risk and explain their journey through the episode of care and the results of which are described in the S5 Text.

 All analyses were conducted in Python version 3.8.6 or R version 4.0.3. Data analysis was performed using scikit-learn 0.24.1 [16] and XGBoost 1.3.3 [23] for gradient descent boosting. SHAP explanations were generated and visualized with SHAP 0.38.1 [11]. Model calibrations generated in R with mgcv 1.8-33 [26] and calibration variances with sandwich 3.0-0 [27]. Source code is available online at: https://github.com/stmilab/DynamicBleeding/. Ethics oversight for data and analyses were provided and approved by the Yale Human Research Protection Program (IRB #0607001639). This study was conducted on a retrospective, de-identified registry data, therefore, no consent was needed or obtained for this work.

### Supporting information

**S1 Fig. Receiver Operating Characteristic Curves for each staged model with a representative fold taken at random for each stage of the model and each 5-fold cross-validation.**
(DOCX)

**S1 Table. Variables included at each stage and their definition as coded in the NCDR data dictionary.** The highlighted rows are those variables that were removed in the multicollinearity analysis.
(DOCX)

**S1 Text. Text describing the multicollinearity experiments.**
(DOCX)

**S2 Fig. Precision Recall Curves for each staged model with a representative fold taken at random for each stage of the model and each 5-fold cross-validation.**
(DOCX)

**S2 Table. Comparison of model performances for bleeding prediction by subgroup on access site.**
(DOCX)

**S2 Text. Text describing the experiments conducted and results of those experiments for comparing femoral vs. radial access results.**
(DOCX)

**S3 Fig. Receiver Operating Characteristic Curves for each staged model with a representative fold taken at random for each stage of the model and each 5-fold cross-validation for the collinearity analysis.**
(DOCX)

**S3 Table. Shift tables following each decision point.** Top value in each cell is number of patients classified into that risk bin by the two respective models. The bottom value in each cell indicates the actual bleeding rate of all patients within that cell. S3 Table shows shift tables before and after each decision, while Table 3 shows a shift table from the

initial to final model. The earlier model is displayed left to right, while the later model is displayed top to bottom. The top number in each cell represents the number of patients assigned to that risk bin by each model. The bottom number in each cell is the overall bleeding rate of all patients in that cell. NaN represents that no patients were in that combination of bins.
(DOCX)

**S3 Text.** **Text that describes the break down of each model/decision and the results, shift tables, and additional results.**
(DOCX)

**S4 Fig.** **Precision Recall Curves for each staged model with a representative fold taken at random for each stage of the model and each 5-fold cross-validation for the collinearity analysis.**
(DOCX)

**S4 Table.** **Demographic information for overall cases, and broken up into those with and without bleeds.**
(DOCX)

**S4 Text.** **Additional description of the variable selection, imputation strategy, and performance metrics selected.**
(DOCX)

**S5 Fig.** **SHAP Tree explainer for Model 2: Access Site.** Procedures performed via femoral access are represented by the narrow red line to the right of the axis. In contrast, the blue points to the left of the axis represent procedures per-formed with radial access, and their elongated shape indicates that the radial access has a variable effect on bleeding risk, with the risk for some procedures being decreased by much more than the risk for others.
(DOCX)

**S5 Text.** **Additional experiments and discussion of case study for changes in risk calculate over stages.**
(DOCX)

**S6 Fig.** **SHAP Tree explainer for Model 3: Cath Lab Visit.**
(DOCX)

**S7 Fig.** **SHAP Tree explainer for Model 4: Pre-Operative Medication Prescription.**
(DOCX)

**S8 Fig.** **SHAP Tree explainer for Model 5: PCI.**
(DOCX)

**S9 Fig.** **SHAP Tree explainer for Model 1: Presentation – of the multicollinearity analysis.**
(DOCX)

**S10 Fig.** **SHAP Tree explainer for Model 2: Access Site of the multicollinearity analysis.**
(DOCX)

**S11 Fig.** **SHAP Tree explainer for Model 3: Cath Lab – of the multicollinearity analysis.**
(DOCX)

**S12 Fig.** **SHAP Tree explainer for Model 4: Medications – of the multicollinearity analysis.**
(DOCX)

**S13 Fig.** **SHAP Tree explainer for Model 5: PCI – of the multicollinearity analysis.**
(DOCX)

**S14 Fig. SHAP Tree explainer for Model 6: Closure – of the multicollinearity analysis.**
(DOCX)

**S15 Fig. Plot of case study risk scores across all model stages.** Case Study A began as high risk but was low risk in the final model. Case Study A did not ultimately bleed. Case Study B began as low risk but was high risk in the final model. Case Study B ultimately experienced a bleed.
(DOCX)

**S16 Fig SHAP explainer for Case Study A.** At each model stage, the prediction is created by summing each variable contribution to risk. Variables on the left (red) are contribute to an increased risk of bleeding, while variables on the right (blue) contribute to a decreased risk of bleeding. Variables are organized such that those providing the strongest change to risk are at the center, with variables providing smaller changes to risk at the outside.
(DOCX)

**S17 Fig. SHAP explainer for Case Study B.**
(DOCX)

## Acknowledgments

The views expressed herein represent those of the authors and do not necessarily represent the official views of the National Cardiovascular Data Registry or its associated professional societies identified online (https://cvquality.acc.org/NCDR-Home).

## Author contributions

**Conceptualization:** Nihar Desai, John Rumsfeld, Sahand Negahban, Harlan Krumholz, Bobak J. Mortazavi.

**Data curation:** Nathan Hurley, Bobak J. Mortazavi.

**Formal analysis:** Bobak J. Mortazavi.

**Investigation:** Bobak J. Mortazavi.

**Methodology:** Nathan Hurley, Nihar Desai, Sanket Dhruva, Rohan Khera, Wade Schulz, Chenxi Huang, Harlan Krumholz, Bobak J. Mortazavi.

**Software:** Nathan Hurley.

**Supervision:** Nihar Desai, Sanket Dhruva, Jeptha Curtis, Frederick Masoudi, John Rumsfeld, Sahand Negahban, Harlan Krumholz.

**Validation:** Nathan Hurley, Rohan Khera.

**Writing – original draft:** Nathan Hurley, Sanket Dhruva, Bobak J. Mortazavi.

**Writing – review & editing:** Nathan Hurley, Nihar Desai, Sanket Dhruva, Rohan Khera, Wade Schulz, Chenxi Huang, Jeptha Curtis, Frederick Masoudi, John Rumsfeld, Sahand Negahban, Harlan Krumholz, Bobak J. Mortazavi.

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
