## [Decision Letter · Decision Letter 0]

Oct 18 2024

PDIG-D-24-00174

Towards a dynamic model to estimate evolving risk of major bleeding after percutaneous coronary intervention

PLOS Digital Health

Dear Dr. Mortazavi,

Thank you for submitting your manuscript to PLOS Digital Health. After careful consideration, we invite you to submit a revised version of the manuscript that addresses the points raised during the review process.

Please submit your revised manuscript within 60 days Oct 18 2024 11:59PM. If you will need more time than this to complete your revisions, please reply to this message or contact the journal office at digitalhealth@plos.org. Please include the following items when submitting your revised manuscript:

We look forward to receiving your revised manuscript.

Kind regards,

Aline Lutz de Araujo

Academic Editor

PLOS Digital Health

Journal Requirements:

1. We do not publish any copyright or trademark symbols that usually accompany proprietary names, eg (R), (C), or TM (e.g. next to drug or reagent names). Please remove all instances of trademark/copyright symbols throughout the text, including ™ on page 20.

Additional Editor Comments (if provided):

The 3 peer reviewers had somewhat different assessments. Collectively, they have identified a number of issues. Many of those involve analytical methods, or at least how those methods are described.

The feature selection process should be described in more detail. It appears that some of the features are highly correlated, which may introduce unnecessary redundancy to the model, as pointed out by Reviewer 1. A data drift issue has also been raised. Although using older datasets does not diminish the merit of the study, the risk of data drift is a limitation worth mentioning. They also requested further explanation on how the model would be implemented in clinical practice.

Reviewer 1:

Nathan Hurley et al. used a large database to build a dynamic model to assess the risk of major bleeding after PCI. In clinical practice, dynamic patient assessment is important, and the authors' approach is fascinating.

However, there are several, analytical and clinical concerns that need to be addressed as follows;

1. The first and most important issue is the definition of the outcome. Major bleeding is defined as occurring after the start of PCI. Since some of the predictors are obtained during PCI, these features no longer seem to be predictive. The outcome should be defined after the predictors are obtained.

2. Hemoglobin was a fairly high predictive variable in the results. First, what makes Hb over 13 a high risk in terms of pathology? What concerns me is the validity of the definition of major bleeding as a decrease in Hb 3 or higher. For example, is it possible that patients with Hb 13 or higher could be dehydrated? The decrease of Hb by fluid infusion is often seen in patients with higher Hb. Since transfusion and intervention/surgery at the bleeding site were also defined as bleeding in this study, it might be helpful to perform sensitivity analyses to confirm that similar results are obtained with the outcome definitions without hemoglobin.

3. It seems to me that the author includes as many variables as possible. For example, insurance does not seem relevant to the assessment of bleeding risk, but is it necessary? If the authors think that these variables should be included, the rationale should be noted.

I am also concerned about the several similar variables(e.g. Hb >13, HB <13, and continuous Hb.).

Furthermore, several variables seem to be the same variable(e.g. FEMALE and Sex, RENFAIL and CKD4).

Please provide reasons why these variables are included.

4. In addition, SHAP is not an analysis that can completely ignore multicollinearity, so it would be better to exclude unnecessary variables.

5. The risk of bleeding is quite different whether the access site is femoral or radial. Is it possible to show the difference in model performance in these subgroups?

Reviewer 2:

Competent study examining the risk of major bleeding for patients undergoing percutaneous coronary intervention, showing the changes in patient risk estimation throughout the course of treatment. They demonstrate the need for more dynamic evaluation of risk estimates, providing potential changes in treatment decision making throughout admissions, accounting for prior treatment decisions made.

Reviewer 3:

The study and manuscript have several areas that could be improved to enhance their overall robustness, applicability, and impact. One significant limitation is that validation was conducted using a temporal split within the dataset, without the inclusion of an external dataset.

To improve the generalizability of the model across different patient populations and healthcare settings, incorporating an external validation cohort is recommended. This could involve using data from a different registry or conducting a prospective study to assess the model’s performance in varied contexts.

Another area for improvement is the integration of real-time data. The study acknowledges the challenge of acquiring real-time data and relies on manually abstracted data from the NCDR.

Future research could explore the integration of real-time data streams from electronic health records (EHRs) to continuously update the model's predictions. This would require developing robust methods for real-time data processing and ensuring the quality of the data being fed into the model.

The study also discusses the need for reducing clinician burden when interacting with the model but does not explore this aspect in depth. Collaborating with clinicians to develop user-friendly interfaces and decision-support tools that seamlessly integrate into existing clinical workflows would be critical. Conducting user experience (UX) studies could ensure that the model's outputs are actionable and easily interpretable by healthcare providers.

While SHAP (SHapley Additive exPlanations) plots are used to interpret the model, the manuscript could provide more detailed explanations of how these insights could be applied in practice. Expanding on how clinicians can use SHAP plots to understand key predictors of bleeding risk and adjust treatment strategies accordingly would make the model more practical. Additionally, providing case studies or examples where the model influenced clinical decisions could help bridge the gap between machine learning outputs and clinical action.

Another consideration is addressing model drift. The model was trained and validated on data up to 2015 and we are in 2024, almost decade later, clinical practices and patient populations evolve over time, potentially leading to model drift. The manuscript could discuss strategies to monitor and update the model regularly to maintain its accuracy over time. This might involve periodic retraining with new data, continuous learning approaches, or deploying a feedback loop where the model’s predictions are compared with actual outcomes to recalibrate the model.

The study does not extensively discuss potential ethical concerns or biases that may arise from the use of machine learning in clinical decision-making. Including a section that addresses the ethical implications of dynamic risk models, such as potential biases (e.g., racial, gender, socioeconomic) and the need for transparent model governance, could enhance the manuscript. Strategies for bias detection and mitigation would help ensure that the model is equitable and fair across different patient subgroups.

The primary outcome considered in the study is any in-hospital bleeding event within 72 hours after PCI. However, the manuscript could be improved by including long-term outcomes such as mortality, re-hospitalization, or recurrent cardiovascular events. This would provide a more comprehensive and practical view of the model’s impact on patient care and could involve exploring how the dynamic risk model correlates with these long-term outcomes.

The discussion in the manuscript touches on the potential utility of the model but could delve deeper into its practical application. Expanding on how the dynamic model could be implemented in different clinical settings, such as tertiary care centers versus community hospitals, and its potential impact on clinical guidelines, would make the research more compelling. Additionally, discussing any cost-benefit analysis or resource implications of implementing such a model could be valuable.

Finally, while the manuscript outlines future research opportunities, it could be more specific. The authors should consider provide concrete examples of how future studies could build on this work—such as by investigating specific subgroups, incorporating additional data types (e.g., genetic, imaging), or exploring alternative machine learning approaches—would give researchers a clearer roadmap for advancing this field.

Reviewers' comments:

Reviewer's Responses to Questions

**Comments to the Author**

1. Does this manuscript meet PLOS Digital Health’s publication criteria ? Is the manuscript technically sound, and do the data support the conclusions? The manuscript must describe methodologically and ethically rigorous research with conclusions that are appropriately drawn based on the data presented.

Reviewer #1: Yes

Reviewer #2: Yes

Reviewer #3: Partly

2. Has the statistical analysis been performed appropriately and rigorously?

Reviewer #1: No

Reviewer #2: Yes

Reviewer #3: Yes

3. Have the authors made all data underlying the findings in their manuscript fully available (please refer to the Data Availability Statement at the start of the manuscript PDF file)?

Reviewer #1: Yes

Reviewer #2: Yes

Reviewer #3: Yes

4. Is the manuscript presented in an intelligible fashion and written in standard English?

PLOS Digital Health does not copyedit accepted manuscripts, so the language in submitted articles must be clear, correct, and unambiguous. Any typographical or grammatical errors should be corrected at revision, so please note any specific errors here.

Reviewer #1: Yes

Reviewer #2: Yes

Reviewer #3: Yes

5. Review Comments to the Author

Please use the space provided to explain your answers to the questions above. You may also include additional comments for the author, including concerns about dual publication, research ethics, or publication ethics. (Please upload your review as an attachment if it exceeds 20,000 characters)

Reviewer #1: Nathan Hurley et al. used a large database to build a dynamic model to assess the risk of major bleeding after PCI. In clinical practice, dynamic patient assessment is important, and the authors' approach is fascinating.

However, there are several, analytical and clinical concerns that need to be addressed as follows;

1. The first and most important issue is the definition of the outcome. Major bleeding is defined as occurring after the start of PCI. Since some of the predictors are obtained during PCI, these features no longer seem to be predictive. The outcome should be defined after the predictors are obtained.

2. Hemoglobin was a fairly high predictive variable in the results. First, what makes Hb over 13 a high risk in terms of pathology? What concerns me is the validity of the definition of major bleeding as a decrease in Hb 3 or higher. For example, is it possible that patients with Hb 13 or higher could be dehydrated? The decrease of Hb by fluid infusion is often seen in patients with higher Hb. Since transfusion and intervention/surgery at the bleeding site were also defined as bleeding in this study, it might be helpful to perform sensitivity analyses to confirm that similar results are obtained with the outcome definitions without hemoglobin.

3. It seems to me that the author includes as many variables as possible. For example, insurance does not seem relevant to the assessment of bleeding risk, but is it necessary? If the authors think that these variables should be included, the rationale should be noted.

I am also concerned about the several similar variables(e.g. Hb >13, HB <13, and continuous Hb.).

Furthermore, several variables seem to be the same variable(e.g. FEMALE and Sex, RENFAIL and CKD4).

Please provide reasons why these variables are included.

4. In addition, SHAP is not an analysis that can completely ignore multicollinearity, so it would be better to exclude unnecessary variables.

5. The risk of bleeding is quite different whether the access site is femoral or radial. Is it possible to show the difference in model performance in these subgroups?

Reviewer #2: Competent study examining the risk of major bleeding for patients undergoing percutaneous coronary intervention, showing the changes in patient risk estimation throughout the course of treatment. They demonstrate the need for more dynamic evaluation of risk estimates, providing potential changes in treatment decision making throughout admissions, accounting for prior treatment decisions made.

Reviewer #3: The study and manuscript have several areas that could be improved to enhance their overall robustness, applicability, and impact. One significant limitation is that validation was conducted using a temporal split within the dataset, without the inclusion of an external dataset. 

To improve the generalizability of the model across different patient populations and healthcare settings, incorporating an external validation cohort is recommended. This could involve using data from a different registry or conducting a prospective study to assess the model’s performance in varied contexts.

Another area for improvement is the integration of real-time data. The study acknowledges the challenge of acquiring real-time data and relies on manually abstracted data from the NCDR. 

Future research could explore the integration of real-time data streams from electronic health records (EHRs) to continuously update the model's predictions. This would require developing robust methods for real-time data processing and ensuring the quality of the data being fed into the model.

The study also discusses the need for reducing clinician burden when interacting with the model but does not explore this aspect in depth. Collaborating with clinicians to develop user-friendly interfaces and decision-support tools that seamlessly integrate into existing clinical workflows would be critical. Conducting user experience (UX) studies could ensure that the model's outputs are actionable and easily interpretable by healthcare providers.

While SHAP (SHapley Additive exPlanations) plots are used to interpret the model, the manuscript could provide more detailed explanations of how these insights could be applied in practice. Expanding on how clinicians can use SHAP plots to understand key predictors of bleeding risk and adjust treatment strategies accordingly would make the model more practical. Additionally, providing case studies or examples where the model influenced clinical decisions could help bridge the gap between machine learning outputs and clinical action.

Another consideration is addressing model drift. The model was trained and validated on data up to 2015 and we are in 2024, almost decade later, clinical practices and patient populations evolve over time, potentially leading to model drift. The manuscript could discuss strategies to monitor and update the model regularly to maintain its accuracy over time. This might involve periodic retraining with new data, continuous learning approaches, or deploying a feedback loop where the model’s predictions are compared with actual outcomes to recalibrate the model.

The study does not extensively discuss potential ethical concerns or biases that may arise from the use of machine learning in clinical decision-making. Including a section that addresses the ethical implications of dynamic risk models, such as potential biases (e.g., racial, gender, socioeconomic) and the need for transparent model governance, could enhance the manuscript. Strategies for bias detection and mitigation would help ensure that the model is equitable and fair across different patient subgroups.

The primary outcome considered in the study is any in-hospital bleeding event within 72 hours after PCI. However, the manuscript could be improved by including long-term outcomes such as mortality, re-hospitalization, or recurrent cardiovascular events. This would provide a more comprehensive and practical view of the model’s impact on patient care and could involve exploring how the dynamic risk model correlates with these long-term outcomes.

The discussion in the manuscript touches on the potential utility of the model but could delve deeper into its practical application. Expanding on how the dynamic model could be implemented in different clinical settings, such as tertiary care centers versus community hospitals, and its potential impact on clinical guidelines, would make the research more compelling. Additionally, discussing any cost-benefit analysis or resource implications of implementing such a model could be valuable.

Finally, while the manuscript outlines future research opportunities, it could be more specific. The authors should consider provide concrete examples of how future studies could build on this work—such as by investigating specific subgroups, incorporating additional data types (e.g., genetic, imaging), or exploring alternative machine learning approaches—would give researchers a clearer roadmap for advancing this field.

6. PLOS authors have the option to publish the peer review history of their article (what does this mean? ). If published, this will include your full peer review and any attached files.

**Do you want your identity to be public for this peer review?** For information about this choice, including consent withdrawal, please see our Privacy Policy .

Reviewer #1: No

Reviewer #2: No

Reviewer #3: No

---

## [Decision Letter · Decision Letter 1]

PDIG-D-24-00174R1Towards a dynamic model to estimate evolving risk of major bleeding after percutaneous coronary interventionPLOS Digital Health Dear Dr. Mortazavi, Thank you for submitting your manuscript to PLOS Digital Health. After careful consideration, we invite you to submit a revised version of the manuscript that addresses the points raised during the review process. Please submit your revised manuscript within 60 days Mar 06 2025 11:59PM. If you will need more time than this to complete your revisions, please reply to this message or contact the journal office at digitalhealth@plos.org. Please include the following items when submitting your revised manuscript:* A rebuttal letter that responds to each point raised by the editor and reviewer(s). You should upload this letter as a separate file labeled 'Response to Reviewers '. This file does not need to include responses to any formatting updates and technical items listed in the 'Journal Requirements' section below.* A marked-up copy of your manuscript that highlights changes made to the original version. You should upload this as a separate file labeled 'Revised Manuscript with Track Changes '.* An unmarked version of your revised paper without tracked changes. You should upload this as a separate file labeled 'Manuscript '. If you would like to make changes to your financial disclosure, competing interests statement, or data availability statement, please make these updates within the submission form at the time of resubmission. Guidelines for resubmitting your figure files are available below the reviewer comments at the end of this letter. We look forward to receiving your revised manuscript. Kind regards, Aline Lutz de AraujoAcademic EditorPLOS Digital Health Leo Anthony CeliEditor-in-ChiefPLOS Digital Healthorcid.org/0000-0001-6712-6626  **Reviewers' Comments:** Reviewer's Responses to Questions

**Comments to the Author**

1. If the authors have adequately addressed your comments raised in a previous round of review and you feel that this manuscript is now acceptable for publication, you may indicate that here to bypass the “Comments to the Author” section, enter your conflict of interest statement in the “Confidential to Editor” section, and submit your "Accept" recommendation.

Reviewer #1: (No Response)

2. Does this manuscript meet PLOS Digital Health’s publication criteria ? Is the manuscript technically sound, and do the data support the conclusions? The manuscript must describe methodologically and ethically rigorous research with conclusions that are appropriately drawn based on the data presented.

Reviewer #1: Partly

3. Has the statistical analysis been performed appropriately and rigorously?

Reviewer #1: No

4. Have the authors made all data underlying the findings in their manuscript fully available (please refer to the Data Availability Statement at the start of the manuscript PDF file)?

Reviewer #1: Yes

5. Is the manuscript presented in an intelligible fashion and written in standard English?

Reviewer #1: Yes

6. Review Comments to the Author

Reviewer #1: The author has addressed many concerns; however, further consideration of multicollinearity is necessary. Specifically, the handling of the following variables may not be appropriate, leading to unstable results and making the SHAP analysis difficult to interpret. It would be advisable to reconsider these aspects.

Firstly, if AGEGT70 and AGELE70 have opposite meanings, retaining both variables will cause multicollinearity. The following binary variables may also raise similar concerns:

・OnsetTimeEst & OnsetTimeNA

・PREHGBGT13 & PREHGBLE13

Notably, in Figure 2, both PREHGBGT13 and PREHGBLE13 appear. Given that these variables have opposite meanings, why do both higher feature values(1=yes?) show higher SHAP values? Generally, if variables have completely opposite meanings, their SHAP values would also be expected to be opposite.

Similarly, for AdmtSource, which has three categories: "ED," "Transfer," and "Other," when decomposed into a one-hot encoding, not creating two variables will result in multicollinearity. The following categorical variables may also raise similar concerns:

・Dominance_1, Dominance_2 & Dominance_3

7. PLOS authors have the option to publish the peer review history of their article (what does this mean? ). If published, this will include your full peer review and any attached files.

**Do you want your identity to be public for this peer review?** For information about this choice, including consent withdrawal, please see our Privacy Policy .

Reviewer #1: No

---

## [Decision Letter · Decision Letter 2]

Towards a dynamic model to estimate evolving risk of major bleeding after percutaneous coronary intervention

PDIG-D-24-00174R2

Dear Dr. Mortazavi,

We are pleased to inform you that your manuscript 'Towards a dynamic model to estimate evolving risk of major bleeding after percutaneous coronary intervention' has been provisionally accepted for publication in PLOS Digital Health.

Best regards,

Aline Lutz de Araujo

Section Editor

PLOS Digital Health

**Additional Editor Comments (if provided):**

**Reviewer Comments (if any, and for reference):**

Reviewer's Responses to Questions

**Comments to the Author**

1. If the authors have adequately addressed your comments raised in a previous round of review and you feel that this manuscript is now acceptable for publication, you may indicate that here to bypass the “Comments to the Author” section, enter your conflict of interest statement in the “Confidential to Editor” section, and submit your "Accept" recommendation.

Reviewer #1: All comments have been addressed

2. Does this manuscript meet PLOS Digital Health’s publication criteria ? Is the manuscript technically sound, and do the data support the conclusions? The manuscript must describe methodologically and ethically rigorous research with conclusions that are appropriately drawn based on the data presented.

Reviewer #1: Yes

3. Has the statistical analysis been performed appropriately and rigorously?

Reviewer #1: Yes

4. Have the authors made all data underlying the findings in their manuscript fully available (please refer to the Data Availability Statement at the start of the manuscript PDF file)?

Reviewer #1: Yes

5. Is the manuscript presented in an intelligible fashion and written in standard English?

Reviewer #1: Yes

6. Review Comments to the Author

Reviewer #1: The authors have addressed all the raised concerns.

7. PLOS authors have the option to publish the peer review history of their article (what does this mean? ). If published, this will include your full peer review and any attached files.

**Do you want your identity to be public for this peer review?** For information about this choice, including consent withdrawal, please see our Privacy Policy .

Reviewer #1: No
